# Impact of COVID-19 Countermeasures on Pediatric Infections

**DOI:** 10.3390/microorganisms10101947

**Published:** 2022-09-30

**Authors:** Naomi Sakon, Tomoko Takahashi, Toshiaki Yoshida, Tatsuya Shirai, Jun Komano

**Affiliations:** 1Osaka Institute of Public Health, Osaka 5370025, Japan; 2Iwate Prefectural Research Institute for Environmental Science and Public Health, Morioka 0200857, Japan; 3Department of Microbiology and Infection Control, Faculty of Fharmacy, Osaka Medical and Pharmaceutical University, Takatsuki 5691041, Japan

**Keywords:** COVID-19, infection control measures, social implementation, pediatric infections, mild lockdown

## Abstract

(1) Background: General infection control measures have been implemented at the societal level against COVID-19 since the middle of 2020, namely, hand hygiene, universal masking, and social distancing. The suppressive effect of the social implementation of general infection control measures on pediatric infections has not been systematically assessed. (2) Methods: We addressed this issue based on publicly available data on 11 pediatric infections reported weekly by sentinel sites in Osaka and Iwate prefectures in Japan since 2010. We obtained the 5-year average for 2015–2019 and compared it to the weekly report for 2020–2021. (3) Results: The rate of 6 of the 11 pediatric infections decreased significantly during 2020–2021, regardless of the magnitude of the prevalence of COVID-19 in both areas. However, only RSV infection, one of the six infections, was endemic in 2021. Exanthem subitum was not as affected by COVID-19 countermeasures as other diseases. (4) Conclusions: The social implementation of infectious disease control measures was effective in controling certain infectious diseases in younger age groups, where compliance with the countermeasures should not be as high as that of adults.

## 1. Introduction

The COVID-19 pandemic has changed people’s ways of life worldwide. COVID-19, initially detected in China in December 2019, quickly spread globally [1,2]. Public health and social measures were taken against COVID-19, including intensive surveillance and social implementation of infection control measures. In 2020 and 2021, the Japanese government declared four one-month states of emergency in response to COVID-19. Unlike city lockdowns, the government *requested* that people avoid going out unnecessarily and work remotely; this was termed a “mild lockdown”. Mass gatherings were restricted to no more than 5000 participants or a capacity of 50%. For example, the 2020 Tokyo Olympics was postponed and held in 2021 using the “Olympic Bubble”. The open hours of restaurants, department stores, and sports facilities were shortened. People were asked to wear a mask at all times, keep a distance from others, and practice hand hygiene when going out. There was no legal penalty for disobedience. These infection control measures were introduced nationwide after the first declaration of a state of emergency against COVID-19 on 7 April 2020. Such countermeasures continued to prevent an explosive increase in COVID-19 cases in Japan until mid-2021. Practicing hand hygiene, wearing masks, and limiting human-to-human contact are key infection control measures in healthcare settings. It is reasonable to speculate that the implementation of these actions at the community level should reduce not only cases of COVID-19 but also cases of other infections if compliance is sufficiently high. We sought to determine the impact of COVID-19 countermeasures conducted at the community level on pediatric infectious diseases. We analyzed the prevalence of 11 diseases before and after the COVID-19 pandemic. The following diseases are reported by sentinel hospitals, mainly for pediatric cases, under the Infectious Disease Surveillance System in Japan based on the Prevention of Infectious Diseases and Medical Care for Patients with Infectious Diseases (hereafter referred to as the “Infectious Diseases Control Law”): influenza, respiratory syncytial virus (RSV) infection, pharyngoconjunctival fever (PCF), group A streptococcal pharyngitis (GASP), infectious gastroenteritis (GE), varicella, herpangina, erythema infectiosum (EI), exanthem subitum (ES), mumps, and hand, foot, and mouth disease (HFMD). Nationwide surveillance data are insufficient to provide the visualization of this occurrence in depth. We focused our analysis on two geographically distant prefectures, Osaka and Iwate in Japan, with different COVID-19 outbreaks. These two areas are distinct in that Osaka is a metropolitan area, and Iwate is situated in the northeast of Japan, with different populations and numbers of COVID-19 patients. Although there have been reports on specific diseases and institutions, such as the low prevalence of influenza during the COVID-19 pandemic [3,4,5,6,7], we targeted pediatric infectious diseases under surveillance to clarify the countermeasures in society. We evaluated the effectiveness and limitations of the infection control measures implemented at the societal level.

## 2. Materials and Methods

### 2.1. Data Collection

Infectious disease surveillance is conducted under the Infectious Diseases Control Law. The 11 diseases analyzed in this work, influenza, RSV infection, PCF, GASP, GE, varicella, herpangina, EI, ES, mumps, and HFMD, are under sentinel surveillance. The number of reported cases per fixed point in a week since 2010 was obtained either from the Osaka or Iwate Prefectural Infectious Disease Information Centers. The features of the targeted diseases are summarized in Table 1. Patient age data were also obtained from the same resources. The number of patients was reported by pediatric clinics and pediatric hospital departments. The number of sentinel hospitals was approximately 200 and 40 in Osaka and Iwate, respectively. For influenza, the numbers of patients were also reported by internal medicine departments, adding approximately 100 and 20 fixed points in Osaka and Iwate, respectively. The number of COVID-19 patients in Osaka and Iwate prefectures was obtained from open data of the Ministry of Health, Labor, and Welfare (https://www.mhlw.go.jp/stf/covid-19/open-data.html, (accessed on 5 April 2022)). Human mobility flow data were obtained from V-RESAS (https://v-resas.go.jp, (accessed on 5 April 2022)), which is provided by the Cabinet Office for Promotion of Regional Revitalization and the Cabinet Secretariat of the headquarters for overcoming population decline and revitalizing the local economy in Japan.

### 2.2. Data Analysis

Data on the weekly number of reports per sentinel site from 2010 to 2021 were used to analyze long-term trends in prevalence. The 5-year average was calculated by the five-week moving average number of reports per sentinel site from 2015 to 2019 that used the data of the 2 weeks before and after the given week. This 5-year average obtained from the above calculations was compared to the number of reports for the relevant week in 2020 and 2021 (Student’s *t*-test). The 53rd week of the year was excluded due to the absence of weeks in some years. The target period was 90 weeks, beginning in the 15th week in 2020, after the first declaration of a COVID-19 state of emergency, and lasting until the 52nd week of 2021. A *p*-value less than 0.001 was considered statistically significant. The epidemiological trend of RSV infections by age since 2016 in Osaka was also assessed.

## 3. Results

### 3.1. General Trends of Pediatric Infections in Osaka and Iwate before the COVID-19 Pandemic

Figure 1 shows epidemic curves for the 11 target infectious diseases in the prefectures of Osaka and Iwate from 2010 to 2021. Before the COVID-19 pandemic, three infections were endemic every year with a distinct peak: herpangina in the warm season and influenza and RSV infection in the cold season. Seasonal trends were also observed for GE, ES, varicella, PCF, and GASP, with several peaks during which these infections were endemic. HFMD was endemic every other year, and EI was endemic every four years during the warm season. Mumps was endemic at 5-year intervals. The epidemic periodicity of EI and mumps was more clearly reported in Osaka than in Iwate. Endemic varicella appeared to be suppressed in comparison to before 2016, with a small seasonal endemic peak. This could be simply due to the vaccination program that was started in 2014. In Japan, the mumps vaccine is optional and not a routine vaccination. A gradual decline in ES was also notable, with an average year-on-year decrease of 1–5% (average of 4%) during the last 10 years. The epidemic curve of GE and herpangina showed a downward trend; on the other hand, influenza, RSV, PCF, and GASP have been on the rise for the past several years. The long-term observation of trends in infectious diseases showed that, when comparing epidemic status, the trends have changed over the past 10 years, and a period of 5 years was considered appropriate for the period covered.

### 3.2. Pediatric Infections during the COVID-19 Pandemic in Osaka and Iwate

#### 3.2.1. Human Mobility Flow and Infectious Diseases during the Mild Lockdown

Osaka and Iwate differed in the magnitude of the COVID-19 pandemic, and therefore, the number of emergency declarations was also different (Figure 2). We evaluated the impact of infection control measures against COVID-19 in both regions under these settings. Human mobility flow was substantially reduced nationwide under the first state of emergency, decreasing by approximately 60% from the figure for the same week in the previous year (Figure A1). Elementary and junior high schools were not closed during the states of emergency subsequent to the first state of emergency, and human mobility was less restricted. In 2021, only weeks 1 and 33 in Osaka and only week 52 in Iwate showed an increase compared with these weeks in 2019. The weeks of the year-end and New Year and summer vacations, which are society-wide vacations that are custom in Japan, corresponded to the weeks of increase. On the other hand, the weeks with an increase in Osaka showed a marked decrease in Iwate. In general, the flow of people during the COVID-19 pandemic period was suppressed compared to the same weeks in 2019, and the social measures remained in place.

#### 3.2.2. Pediatric Infections during the COVID-19 Pandemic and Their Comparison with the 5-Week Moving Average over the Past 5 Years 

In 2020, no influenza epidemic peaks were observed, and cases of PCF, GASP, GE, and herpangina were reduced, although sporadic cases were reported (Figure 1) in Osaka and Iwate. Cases of RSV in 2020 showed a marked decrease in Osaka; however, the number of cases increased at the end of the year in Iwate. Cases of ES did not show a downward trend, and an increase in the number of patients was observed in Iwate. In 2021, influenza epidemic peaks were not observed in either region, and the number of GASP outbreaks continued to decline. The number of ES infections was lower in 2021 than that in 2020. The number of RSV cases increased considerably in both Osaka and Iwate in 2021. In Iwate, as in previous years, the outbreak cases peaked, then fell, but then increased again. Based on the periodicity of transmission, HFMD was predicted to be nonendemic in both regions in 2020, and although 2021 was an epidemic year based on periodicity, the peak values in Osaka and Iwate were 4.27 in weeks 44 and 35, respectively. These values were the lowest among the five epidemic peaks in the past 10 years. For EI and mumps, periodicity also inferred no epidemics in 2020 and 2021. HFMD was also considered to be nonendemic in 2020 based on periodicity. Vaccines were available for varicella and mumps, and large outbreaks were already under control (Figure 1). Based on the above background, four diseases, HFMD, EI, varicella, and mumps, were excluded from the evaluation of the impact of COVID-19 countermeasures on infectious disease control. The rotavirus vaccine was introduced in Japan in November 2011, but routine vaccination was not available until October 2020. Infectious gastroenteritis may be caused by other pathogens, such as norovirus, and was therefore included in the analysis. Ultimately, seven diseases were included in the analysis, and the 5-year average was compared to the number of reports from weeks 15 to week 52 in 2020 and 52 weeks in 2021 (Student’s *t*-test, *p* < 0.001) (Table 2). No significant differences were observed in the decline of influenza in Osaka in 2020, but no epidemics were observed in either region for the 2020/2021 season. The four infections, PCF, GASP, GE, and herpangina, decreased significantly over the two years. On the other hand, there was a notable difference in RSV infection that significantly decreased in 2020, but the number of reported cases increased in 2021 in both prefectures. In Iwate, ES significantly increased, unlike the declining trend in pediatric disease; ES was less susceptible to the infectious disease control measures implemented by society.

### 3.3. Analysis of RSV Infections by Age Group

We analyzed the age distribution of cases of RSV infection. In Osaka, most patients were one year old during the period from 2016 to 2019. In 2020, there was a sharp decrease in patients in all age groups, but in 2021, the number of patients increased in the age group from 1 to 5 years (Figure 3). These data suggested that RSV infection was indeed suppressed in 2020, resulting in an increase in RSV-susceptible infants. However, the level of difference in 2020 was not significant when compared to the previous five years (Table 2). RSV infections spread rapidly in Osaka and Iwate, although human flow mobility remained reduced during 2020 and 2021 relative to 2019 (Figure A1).

## 4. Discussion

Until effective vaccines were available, the basic strategies against COVID-19 were hand washing, mask-wearing, the avoidance of closed rooms or crowds, and the restriction of human mobility. This situation constituted a social experiment suitable for assessing whether implementing such measures at the community level would be effective against community-acquired infections. Critical assessment of the current status of infections has the potential to aid in the development of effective countermeasures against emerging and reemerging infectious diseases in the future. Therefore, we focused on the impact of COVID-19 countermeasures on diseases that have been under surveillance regimens for a long period of time, such as pediatric infections. However, it has been reported that cases of notifiable infectious diseases such as influenza and norovirus decreased in some countries during the COVID-19 pandemic period [3,4,5,6,7]. These reports do not necessarily verify the overall effect of these countermeasures on infectious diseases.

We found that at least 6 out of the 11 targeted pediatric infections were controlled. This is the first clear evidence that public health interventions are effective against a wide variety of pediatric infections. These infections are transmitted either via droplets or direct contact, or both. Infections of infants and school-age children should be discussed separately because the former is unable to comply with infection control requirements. In Japan, childcare facilities were not entirely closed under the state of emergency. Infants did not wear masks at nursery schools because choking while wearing masks is considered a higher risk to their health than COVID-19. Moreover, infants cannot properly practice other COVID-19 countermeasures. The question arises of why pediatric infections were controlled in 2020. School closures, masks, and thorough hand sanitization were observed among schoolchildren, and the decrease in diseases that are more prevalent among schoolchildren has been sustained to this day. In addition, it is assumed that compliance with infectious disease control measures by adults played an important role in diseases that affected all age groups. Influenza and norovirus infections are good examples. Approximately 55% (39.9–65.5%) of foodborne GE outbreaks have been attributed to norovirus in the past 10 years. Only a few foodborne norovirus outbreaks were reported in 2020–2021 because restaurant use was limited, especially during a state of emergency (Table A1). The adult population was removed from sites of norovirus infection. Therefore, adult-to-child transmission was prevented, resulting in no GE outbreaks at nursery schools. The prevalence of norovirus GII.4, the most prevalent genotype worldwide, was low during the COVID-19 pandemic. On the other hand, RSV infections, for which 1- to 2-year-olds are the target age for infection and disease onset, were prevalent. The accumulation of susceptible persons and the fact that the target age group had difficulties implementing infection control measures are presumed to be the reasons for the rapid increase in the number of cases. We also suggest that this was due to the way the pathogens spread. The main transmission route of RSV is via droplets and contact. However, aerosol infection has been suggested as a transmission route for RSV [8]. The reproduction coefficient for RSV infection was estimated to be 0.92–1.76 in a cohort study in the Philippines [9] and 3.0 in the U.S. [10], which is higher than that of 1.28–1.30 for influenza [11,12].

One caveat when making inferences from the data was that the healthcare system was almost saturated in 2020 due to the COVID-19 pandemic. It is possible that the low prevalence of infections could be due to patients’ reluctance to visit medical facilities. In Iwate, the number of ES patients increased and was thought to be partly because patients spent more time at home due to the suppression of social activities. Refraining from medical examinations did not affect the number of ES patients. For the same reason, an increase in the number of ES patients was expected in Osaka, but a slight decrease was observed. This is probably due in no small part to patients refraining from receiving medical examinations in areas with endemic COVID-19. However, RSV epidemics were also detected, and the period of withholding was considered to be limited.

This study demonstrates that the social enforcement of hand hygiene, universal mask-wearing, and avoidance of crowds and travel in the form of a request is effective in curbing outbreaks of contagious pediatric diseases. Mobility flow does not necessarily serve as an indicator of infection control. The adult population appears to play a major role in the control of pediatric infections. It is concerning that once vaccines and other interventions for COVID-19 are implemented, the implementation of general infection control measures will be weakened, and pediatric infections may begin to increase and adversely impact society.

## 5. Conclusions

The society-wide implementation of COVID-19 countermeasures significantly contributed to the reduction in pediatric infections over a two-year period. However, RSV and ES infections were the only exceptions. It remains to be seen which countermeasure(s) had the greatest effect on the control of pediatric infections. It is concerning that once COVID-19 is managed by vaccines and other interventions, pediatric infections will begin to increase and impact our lives. 

## References

## Figures and Tables

**Figure 1 microorganisms-10-01947-f001:**
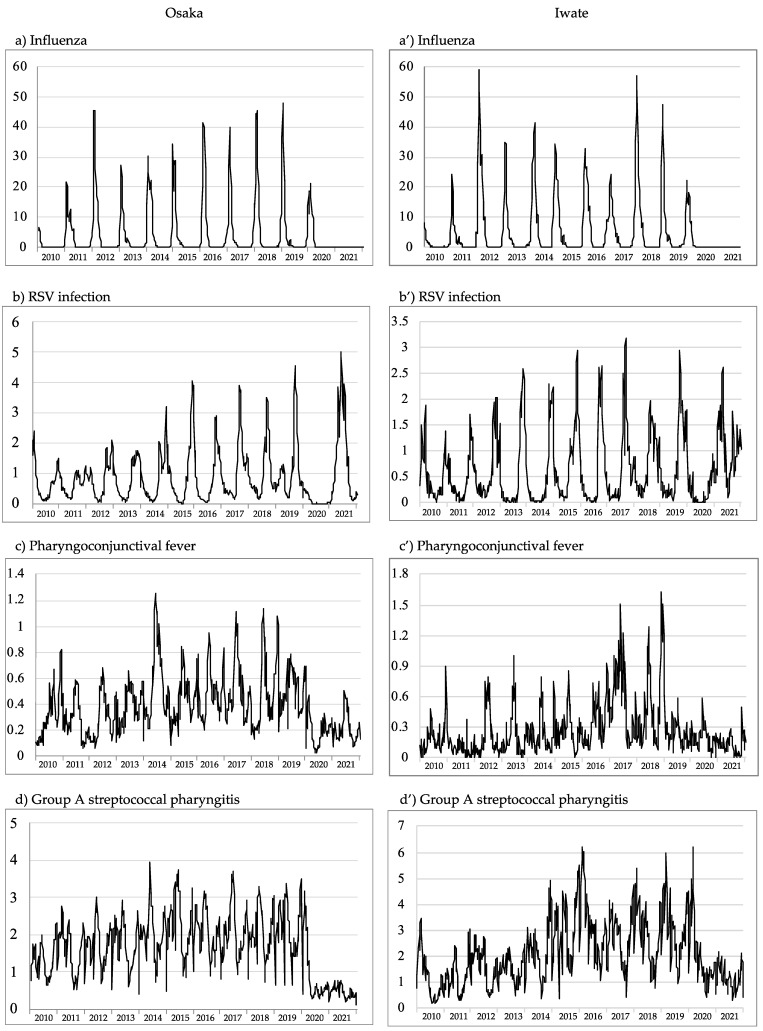
Trends in weekly reported cases of infectious diseases per sentinel site from 2010 to 2021 in Osaka and Iwate prefectures. The 11 diseases were reported mainly among pediatric patients, with data from both prefectures: (**a**–**k**) in Osaka and (**a’**–**k’**) in Iwate.

**Figure 2 microorganisms-10-01947-f002:**
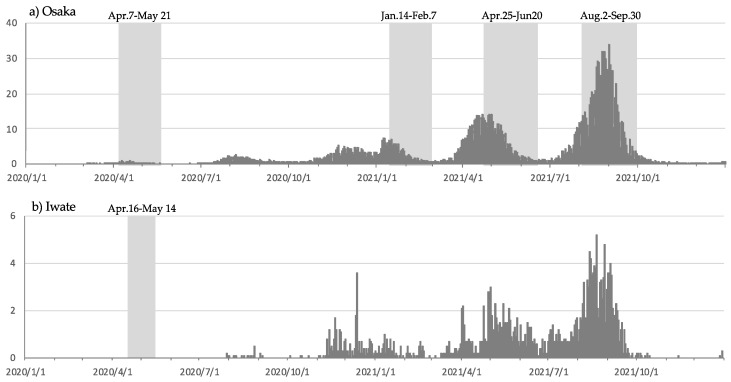
The daily reported number of confirmed cases of SARS coronavirus 2 in distinct regions in Japan from 16 January 2020 to 2021: (**a**) in Osaka, (**b**) in Iwate. These data were taken from the open resources of the Japan Broadcasting Corporation. The period of the declared state of emergency is indicated by gray shading and the date. Osaka is one of the endemic areas in Japan. There were no reports of confirmed COVID-19 cases until the end of July 2020 in Iwate.

**Figure 3 microorganisms-10-01947-f003:**
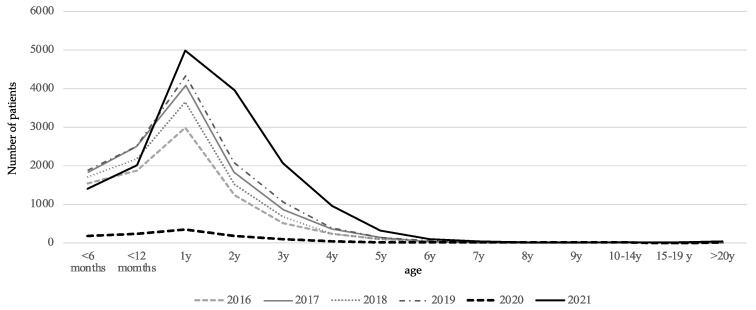
Age distribution of RSV infection in Osaka from 2016 to 2021. The annual numbers of reported patients were 8542, 11,675, 10,174, 12,478, 1129, and 15,915 for each year from 2016 to 2021, respectively.

**Table 1 microorganisms-10-01947-t001:** Summary of the 11 infectious diseases reported by sentinel hospitals in Japan.

	Influenza	RSV Infection (RSV)	Pharyngoconjunc-Tival Fever (PCF)	Group A Streptococcal Pharyngitis (GASP)	Infectious Gastroenteritis (GE)	Herpangina	Exanthem Subitem (ES)	Hand, Foot and Mouth Disease (HFMD)	Erythema Infectiosum (EI)	Varicella	Mumps
pathogen	influenza virus	Respiratory syncytial virus	Adenovirus	*Streptococcus pyogenes*	Rotavirus A, Norovirus etc.	Coxsackie-virus A	Human herpes virus 6,	Enterovirus A (mainly CA16, EV71)	Human parvovirus B19	varicella zoster virus	Mumps virus
			(mainly 3,4,7)				Human herpes virus 7				
transmission	droplet or contact infection	droplet or contact infection	droplet or contact infection	droplet or contact infection	fecol-oral, contact infection, foodborn	fecol-oral, respiratory route	droplet or contact infection	fecol-oral, respiratory route	droplet or contact infection	droplet or contact infection, airborn	via respiratory droplets and saliva
target population of the surveillance	All populations	<15 years old	<15 years old	<15 years old	<15 years old (mainly)	<15 years old	<15 years old	<15 years old	<15 years old	<15 years old	<15 years old
predominant age	<10	<1 years old	<5 years old (60%)	5 to 9 years old	Not particular	<5 years old (90%)	5 to 9 yars old	<5 years old (90%)	<1 years old	<9 years old	5 to 9 yars old
seasonality in Japan	January–Februart	November-January	July–August	April-July & October–December	November–January (Norovirus),	July–August	June–July	July	none	December–July	none
					April (Rotavirus)						
periodicity	every year	every year	every year	every year	every year	every year	every 3–4 years	every other year	none	every year	every 5–6 years
vaccination	routine: elderly				routine: Rotavirus A vaccine from October 2020					routine: to 1 years old from October 2014	voluntary

**Table 2 microorganisms-10-01947-t002:** Test of 5-week moving averages over the past 5 years and the number of patients reported during the COVID-19 pandemic.

	**2020 (15–52 w)**		**2021**	
	**Osaka**	**Iwate**	**Osaka**	**Iwate**
Influenza	0.0053	*p*< 0.001	*p* < 0.001	*p* < 0.001
RSV infection	*p* < 0.001	*p* < 0.001	0.02061	0.01278
Pharyngoconjunctival fever	*p* < 0.001	*p* < 0.001	*p* < 0.001	*p* < 0.001
Group A streptococcal pharyngitis	*p* < 0.001	*p* < 0.001	*p* < 0.001	*p* < 0.001
Infectious gastroenteriris	*p* < 0.001	*p* < 0.001	*p* < 0.001	*p* < 0.001
Herpangina	*p* < 0.001	*p* < 0.001	0.0052	*p* < 0.001
Exanthem subitum	0.002433	*p* < 0.001 *	*p* < 0.001	0.19267

*p* < 0.001 *: The number of ES cases in Iwate significantly increased.

## Data Availability

Not applicable.

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
