# Peer review of "Impact of COVID-19 Countermeasures on Pediatric Infections"

_microorganisms, 2022, doi:10.3390/microorganisms10101947_

Round 1

Reviewer 1 Report

1.      Line 29- define mass gatherings in terms of number of people, space available and time limitations, if any.

2.      Line 32- Re-phrase the sentence.

3.      Line 45- please explain why these diseases were selected.

4.      Line 73- spell check the title of paragraph

5.      Line 244-247- sentences are repeated from previous paragraph.  Re-phrase them to imply the same meaning.

Author Response

1.Line 29- define mass gatherings in terms of number of people, space available and time limitations, if any.

 Response 1: Thank you for the comment. This is important information, and we have included it in Lines 29-31.

2.Line 32- Re-phrase the sentence.

    Response 2: We changed the sentence as follows: There was no legal penalty for disobedience.

3.Line 45- please explain why these diseases were selected.

   Response 3: Thank you for the comment. In the introduction, it is mentioned that 7 diseases were selected for analysis. However, we decided that this was not appropriate for the introduction section because the diseases were selected based on the analysis of epidemic curves before the COVID-19 pandemic (Fig. 1). The text after Line 42 has been edited.

4.Line 73- spell check the title of paragraph

      Response 4: The spelling errors have been corrected.

5. Line 244-247- sentences are repeated from previous paragraph.  Re-phrase them to imply the same meaning.

    Response 5: Thank you for this suggestion. We have rewritten the phrase as follows: The society-wide implementation of COVID-19 countermeasures significantly contributed to the reduction in pediatric infections over a two-year period. However, RSV and ES infections were the only exceptions.

Reviewer 2 Report

The manuscript by Naomi Sakon and collogues, “ Impact of COVID-19 countermeasures on pediatric infections” aims to study the impact of social implementation during the COVID-19 pandemic on pediatric infections by comparing the pediatric infections data in two areas in Japan. The topic is interesting. The authors pointed out interesting conclusions based on their analysis. However, the manuscript is not organized properly: the presentation of results is unclear; details(for example, statistical methods) are essential in the methods and the materials part are missing. In addition, all the countermeasures for COVID-19 were lifted in March 2022 in Japan, strongly suggest to add the data analysis of pediatric infections after March 2022 if the data are available.

Major concerns

1.    The organization and description of results part in this manuscript make it hard to understand what the authors would like to point out. Pediatric infection trends before and after COVID-19 are suggested to be described together. It is not very clear what the standard is to exclude diseases from the evaluation of the impact of COVID-19-social enforcement on pediatric diseases. What is the logical connection for analysis of RSV infections by age group? Many sentences are difficult to understand. For example, lines 126-127 (which year are talking about) and line 129(13 weeks later, compared to what time?). Please carefully organize and describe the results part.

2.    Data analysis should be described properly. What statistical methods were used in this research to calculate the p-value for different data?

Minor concerns

Line 11.  ”(2)” was used before Methods. Where is the (1), should be put before Background?  Please also check line 19 about a similar situation.

Line 13-14. Suggest to point out what method(s) were utilized to compare the incidence and what kind of data was collected.

Line 14-15. A more detailed results description is required here. For example, what infections were decreased and in what magnitude? Which one is the most (or less) affected?

Line 41. Seven of the following …? What is the reason for that  “seven” was selected?

Line 45. What is the connection of “ infectious diseases spread from one domestic region to another” with the sentences around?

Line 46-47. It is not very clear why chose two distinct areas (not different areas) in order to understand the surveillance comprehensively.

Line 57. What is the law? Please mention it here.

Line 79. Please provide the reason for “ The 53rd week of the year was excluded.”

Line 190. Is it “ focused on the impact of COVID-19 on …” or focused on the impact of COVID-19 countermeasures on…”

Line 197. Is the description “ This is the first clear evidence that” acuate?

Please check the language of the whole manuscript  carefully   

For example

Line 17 should be “where compliance with the countermeasures…”

Line 23 should be “The COVID-19 pandemic has…”

Line27. “Unlike with” should be “Unlike”

Line28, is the word “requested” needed to be italic?

Line 48-49. Please rewrite the sentence.

Line 54. Should be “at the societal level”

Line 73 should be “Data analysis”

Author Response

Thank you for reviewing our manuscript. I am appreciate for many important comments for making our paper improved.
We thought it would be possible to verify the impact of COVID-19 countermeasures on pediatric infectious diseases because the identification of pediatric infectious diseases has been systematized over a long period of time.
  We agree with you and have incorporated this suggestion throughout our paper. Apologies for the many English mistakes. We submitted them to English editing service and paid close attention to them.  

Round 2

Reviewer 2 Report

 Thank you for the response letter,

I am satisfied with the auditor's  responses. I do not have additional comments.

Best Regards